

# An exploration of strategies used by dressage horses to control moments around the center of mass when performing passage

Hilary M. Clayton[1,*] and Sarah Jane Hobbs[2,*]

[1] Department of Large Animal Clinical Sciences, Michigan State University, East Lansing, MI, United States of America
[2] Centre for Applied Sport and Exercise Sciences, University of Central Lancashire, Preston, Lancashire, United Kingdom
[*] These authors contributed equally to this work.

## ABSTRACT

**Background**. Locomotion results from the generation of ground reaction forces (GRF) that cause translations of the center of mass (COM) and generate moments that rotate the body around the COM. The trot is a diagonally-synchronized gait performed by horses at intermediate locomotor speeds. Passage is a variant of the trot performed by highly-trained dressage horses. It is distinguished from trot by having a slow speed of progression combined with great animation of the limbs in the swing phase. The slow speed of passage challenges the horse's ability to control the sagittal-plane moments around the COM. Footfall patterns and peak GRF are known to differ between passage and trot, but their effects on balance management, which we define here as the ability to control nose-up/nose-down pitching moments around the horse's COM to maintain a state of equilibrium, are not known. The objective was to investigate which biomechanical variables influence pitching moments around the COM in passage.
**Methods**. Three highly-trained dressage horses were captured by a 10-camera motion analysis system (120 Hz) as they were ridden in passage over four force platforms (960 Hz). A full-body marker set was used to track the horse's COM and measure balance variables including total body center of pressure (COP), pitching moments, diagonal dissociation timing, peak force production, limb protraction–retraction, and trunk posture. A total of twenty passage steps were extracted and partial correlation (accounting for horse) was used to investigate significant ($P < 0.05$) relationships between variables.
**Results**. Hindlimb mean protraction–retraction correlated significantly with peak hindlimb propulsive forces ($R = 0.821$; $P < 0.01$), mean pitching moments ($R = 0.546$, $P = 0.016$), trunk range of motion, COM craniocaudal location and diagonal dissociation time ($P < 0.05$).
**Discussion**. Pitching moments around the COM were controlled by a combination of kinematic and kinetic adjustments that involve coordinated changes in GRF magnitudes, GRF distribution between the diagonal limb pairs, and the moment arms of the vertical GRFs. The moment arms depend on hoof placements relative to the COM, which were adjusted by changing limb protraction–retraction angles. Nose-up pitching moments could also be increased by providing a larger hindlimb propulsive GRF.

Corresponding author
Hilary M. Clayton,
claytonh@cvm.msu.edu

## INTRODUCTION

Horses are cursorial animals capable of performing a wide repertoire of gaits over a large range of speeds. In the past, the horse's locomotor prowess was exploited in transportation, warfare and agriculture but is now primarily important in equestrian sports. The growth of equestrian sports has led to an upsurge of interest in the physiological and biomechanical characteristics of the performance of equine athletes. Dressage is an Olympic equestrian sport in which horses perform natural and artificial gaits in a variety of patterns to demonstrate an advanced level of control of locomotor kinematics and kinetics. Highly-trained dressage horses offer a unique opportunity to study the mechanics of equine locomotor performance and balance control.

During locomotion, the limbs generate ground reaction forces (GRF) that translate the center of mass (COM) and create turning forces (moments) around the COM. Sagittal plane moments around the COM cause pitching rotations of the horse's body in a nose-up or nose-down direction. When traveling at constant speed, the sagittal plane moments within a stride fluctuate as the fore-aft distance between the COM and body center of pressure (COP) changes (*Lee, Bertram & Todhunter, 1999*; *Biewener et al., 2009*; *Hobbs, Bertram & Clayton, 2016*). In quadrupeds locomotor function is related to posture (*Alexander, 1984*). Trunk and neck postural stability and synchrony are important in trotting horses, as this helps to maintain the vestibular apparatus orientation relative to gravity (*Dunbar et al., 2008*), so pitching moments are controlled to ensure that their effect is small. Roll and yaw moments may also destabilize the body and need to be managed, but in dogs and goats sagittal plane pitching moments were found to be larger (*Biewener et al., 2009*). For cursorial mammals with vertically oriented limbs and a high COM, the ability to balance pitching moments in the direction of motion is therefore important (*Hobbs, Bertram & Clayton, 2016*).

In horses trotting at moderate speed, three fundamental motor control strategies are used to balance sagittal-plane pitching moments (*Hobbs, Bertram & Clayton, 2016*). The first motor control strategy is temporal dissociation of the diagonal limbs at landing (diagonal dissociation), the second motor control strategy is adjustment of craniocaudal hoof contact position relative to the COM by changing limb protraction–retraction, and the third motor control strategy is alteration of the vertical force distribution between concurrently-loaded fore- and hindlimbs.

Dressage horses are trained to maintain the trunk in a more nose-up posture and to control pitching rotation around the COM. In equestrian terminology, this is described as being in an uphill balance and is one of the components of collection and self-carriage (*Holmström, Fredricson & Drevemo, 1995*). One of the most challenging movements

performed by dressage horses is a slow, majestic type of trot called passage, which demonstrates the highest degree of collection, elevation of the forehand, cadence and suspension in the trot (*Fédération Equestre Internationale, 2014*). These requirements present a challenge with regard to managing pitching moments and some dressage horses fail to reach the highest levels of competition due to an inability to learn the skills necessary to stabilize their trunk in passage.

Although passage is defined as a two-beat gait, hind-first diagonal dissociation occurs consistently on landing (*Holmström, Fredricson & Drevemo, 1994*; *Clayton, 1997*; *Weishaupt et al., 2009*) and this has been shown to influence pitching moments at trot (*Hobbs, Bertram & Clayton, 2016*). Postural characteristics of passage that distinguish it from trot include reduced ranges of limb protraction–retraction (*Holmström, Fredricson & Drevemo, 1995*; *Weishaupt et al., 2009*). Reduced limb protraction–retraction affects the hoof contact positions relative to the proximal limb segments and to the COM location throughout stance. Also, all limbs generate higher vertical impulses in passage than in trot with a relatively greater increase in the hindlimbs compared with the forelimbs (*Holmström, Fredricson & Drevemo, 1995*; *Clayton, Schamhardt & Hobbs, 2017*). These findings suggest that horses may use all three of the motor control strategies described for the trot to manage their posture and control pitching moments when performing the technically difficult movement of passage.

This study was designed to explore the intricacies of controlling sagittal plane trunk orientation and rotation in passage, which will lead to a better understanding of the mechanisms available to quadrupeds for managing moments around the COM. On a practical level, the insights gained will elucidate understanding the training challenges and inherent physical limitations that make it difficult for some horses to perform passage. The specific aim was to identify biomechanical variables that affect control of moments around the COM when performing passage. We predict that all three motor control strategies, which involve manipulation of temporal kinematics, linear kinematics and GRFs, will have a demonstrable effect on the horse's posture, represented in the sagittal plane by the orientation of the trunk, and the moments around the COM. Thus, the experimental hypotheses are that diagonal dissociation, limb protraction–retraction, and fore:hind vertical force distribution affect trunk inclination and pitching moments around the horse's COM during passage.

## MATERIALS AND METHODS

The study was performed with approval from the Michigan State University Institutional Animal Care and Use Committee under protocol number 02/08-020-00.

### Experimental data collection

Three highly-trained Lusitano dressage horses; mass 607 ± 9 kg ridden by the same highly experienced rider; mass 61.5 kg were used for the study. All horses were judged by an experienced veterinarian to be free from lameness when trotting in a straight line. They were accustomed to the laboratory environment before data collection commenced.

Retro-reflective markers secured to the skin were tracked at 120 Hz using a 10-camera Motion Analysis System (Motion Analysis Corporation, Santa Rosa, California, USA) to acquire a full body kinematic model of the horse as described in *Hobbs, Richards & Clayton (2014)* with the omission of trunk tracking markers T10–T18. The horses warmed up in a riding arena prior to data collection. Once suitably prepared they performed a series of trials of passage along a runway in which a series of four force plates (Bertec Corporation, Columbus, Ohio, USA) recording at 960 Hz were embedded. Successful trials were those in which the horses moved straight and consistently through the data collection volume with only one hoof at a time being in contact with each force plate.

The timings of hoof contacts and lift offs were identified from the force data using a threshold of 50 N. Diagonal steps for the left forelimb and right hindlimb pair (LFRH) and the right forelimb and left hindlimb pair (RFLH) were extracted when both limbs of the diagonal pair made valid contacts with different force plates. Summed fore- and hindlimb GRF for each diagonal step were calculated and the time of zero summed longitudinal force (Tzero) was used to separate braking and propulsive phases.

Variables of interest were chosen based on those identified by *Hobbs, Bertram & Clayton (2016)* as being important to the management of pitching moments during trotting. Craniocaudal COM location relative to the diagonal hoof placements was calculated from the distance by which the COM was behind (caudal to) the grounded fore hoof divided by the distance between the diagonal fore and hind hooves. Body center of pressure (COP) location was determined based on the magnitudes of the vertical forces in the fore- and hindlimbs combined with the COP locations of concurrently-loaded hooves. The instantaneous location of the body COP was then expressed relative to the base of support as the relative distance from the position of the grounded fore hoof divided by the distance between the diagonal fore and hind hooves. The body's COM and COP locations were therefore reported as ratios with higher values indicating greater proximity to the hind hoof.

Moments about the COM due to the effect of GRFs (MGRF) (Nm/kg) were determined over time for each diagonal step (*Hobbs, Richards & Clayton, 2014*) and mean values were calculated separately for braking and propulsive phases with positive values representing a nose-down moment. Peak vertical force (GRFV) (N/kg) and the time taken to reach peak force ($T$) (% stance) were measured for each limb.

Diagonal dissociation time (DIS) (s) was calculated as the time elapsing between hind and fore contacts of each diagonal pair with the value of hind-first contacts being designated positive and fore-first contacts being designated negative. Limb protraction and retraction angles were measured relative to the vertical by representing the forelimb/hindlimb as a line from the tuber spinae scapulae/greater femoral trochanter to the center of rotation of the distal interphalanegeal joint. Protraction was negative, retraction was positive. Mean protraction–retraction angles over the entire stance phase were calculated for the fore ($P$-$R_F$) and hind ($P$-$R_H$) limbs.

Trunk orientation was represented by a line from the cervical C6/C7 junction to the second coccygeal vertebra (CA2). Trunk inclination was the angle between that line and a horizontal line through the C6/C7 junction. Positive values indicated CA2 was higher than

C6/C7 junction. The range of trunk angular motion ($ROM_T$) and mean trunk inclination ($INC_T$) during each diagonal step (degrees) were calculated to represent measures of dynamic posture, as described by *Hobbs, Bertram & Clayton (2016)*. All variables of interest were calculated in Visual 3D (C-Motion Inc., Germantown, MD, USA).

### Data analysis

Tabulated data were imported into SPSS (IBM Corporation, Armonk, New York, USA) for analysis. Data were tested for normality using a Kolmogorov–Smirnov test and all variables were found to be normally distributed except mean forelimb protraction–retraction angle, which was log transformed. Partial correlations (*Morrison, 1976*) controlling for horse were used to determine the relationships between variables and to evaluate balance strategies where significant relationships existed. Significance was set at $P < 0.05$.

## RESULTS

All variables of interest were pooled, which provided a total of 20 steps from the 3 horses, (10 steps from each diagonal pair). Mean and standard deviation (s.d.) are reported for each variable for the pooled data in Table 1, together with the correlations between variables, significance levels and a subjective classification of the variables into the three motor control strategies previously identified (*Hobbs, Bertram & Clayton, 2016*). Standing COP location was (mean ± s.d.) $0.4 \pm 0.00$ (ratio), indicating that the fore:hind vertical force distribution ratio was 60:40. Standing trunk inclination was $11.9 \pm 0.01$ degrees for the three horses.

The COM moved forward at an average speed of $1.22 \pm 0.18$ m/s. Relative to the grounded limbs it was positioned at approximately 70% of the distance from forelimbs to hindlimbs at the start of stance then progressed forward to a position 20–40% of diagonal distance behind the forelimb at lift off. Greater mean hindlimb protraction, which indicates use of the second motor control strategy, placed the hind hoof significantly ($R = -0.771$; $P < .01$) closer to the COM and significantly ($R = 0.546$; $P < .05$) increased nose up moments during propulsion (Table 1).

The hind hoof was predominantly the first of the diagonal pair to contact the ground, and when this occurred the COP initially coincided with the hind hoof position, then moved forward gradually until it coincided with the position of the fore hoof, which was always the last hoof to lift off. Through most of stance the COP tracked the COM quite closely (Fig. 1), which was achieved by adjusting the relative distribution of the vertical GRF between the fore- and hindlimbs; as the COP moved closer to the fore hoof, the forelimb supported a greater proportion of the total vertical GRF.

Trunk inclination had strong negative correlations with the positions of both the COM and COP (Table 1), such that greater elevation of the forehand was associated with closer proximity of the COM and COP to the hind hoof.

The moments around the COM due to the vertical and longitudinal GRFs tended to act in opposite directions through much of stance, which had the effect of keeping the total moments around the COM relatively small (Fig. 2). The net moment was nose-down in early diagonal stance and nose-up in late stance. Post hoc correlations were performed

**Table 1 Mean and (standard deviation) for pooled data of 20 steps of passage and relationship between variables (accounting for horse).** Shaded boxes indicate the balance strategy classifications of the variables.

| Variable | Mean (s.d.) | MGRF Br | MGRF Pr | $ROM_T$ | $INC_T$ | COM | DIS | $T_F$ | $T_H$ | $P\text{-}R_F$ | $P\text{-}R_H$ | COP | $GRFV_F$ | $GRFV_H$ |
|---|---|---|---|---|---|---|---|---|---|---|---|---|---|---|
| MGRF Br (Nm/kg) | 0.39 (0.32) | 1 | | | | | | | | | | | | |
| MGRF Pr (Nm/kg) | −0.19 (0.32) | −0.317 | 1 | | | | | | | | | | | |
| $ROM_T$ (deg) | 2.71 (1.15) | 0.237 | −0.754** | 1 | | | | | | | | | | |
| $INC_T$ (deg) | 8.03 (1.34) | 0.082 | −0.066 | 0.000 | 1 | | | | | | | | | |
| COM (ratio) | 0.42 (0.03) | −0.266 | −0.272 | 0.345 | −0.634** | 1 | | | | | | | | |
| DIS (s) | 0.03 (0.03) | 0.125 | −0.263 | 0.532* | −0.529* | 0.481* | 1 | | | | | | | |
| $T_F$ (% stance) | 0.46 (0.06) | −0.040 | 0.097 | −0.134 | 0.183 | −0.515* | −0.468* | 1 | | | | | | |
| $T_H$ (% stance) | 0.40 (0.06) | −0.156 | −0.080 | −0.180 | 0.203 | 0.045 | −0.332 | −0.074 | 1 | | | | | |
| $P\text{-}R_F$ (deg) | −6.16 (1.81) | 0.289 | −0.019 | 0.143 | 0.538* | −0.385 | −0.315 | −0.016 | 0.208 | 1 | | | | |
| $P\text{-}R_H$ (deg) | −1.57 (2.26) | 0.037 | 0.546* | −0.729** | 0.299 | −0.771** | −0.515* | 0.355 | −0.033 | 0.104 | 1 | | | |
| COP (ratio) | 0.43 (0.04) | −0.085 | 0.068 | 0.011 | −0.650** | 0.506* | 0.440 | −0.196 | −0.001 | −0.444 | −0.238 | 1 | | |
| $GRFV_F$ (N/kg) | 9.56 (0.94) | 0.182 | 0.348 | −0.125 | 0.186 | 0.071 | −0.056 | −0.496 | −0.042 | 0.410 | 0.025 | 0.002 | 1 | |
| $GRFV_H$ (N/kg) | 7.68 (0.54) | 0.412 | −0.274 | 0.316 | 0.331 | −0.136 | −0.094 | −0.138 | −0.033 | 0.550* | 0.032 | −0.393 | 0.428 | 1 |

**Key:**

| | |
|---|---|
| ▓ | Balance strategy 1: Relative fore-aft contact timing |
| ▒ | Balance strategy 2: Foot contact position |
| ░ | Balance strategy 3: Fore-aft vertical force distribution |

**Notes.**

*$P < .05$.

**$P < .01$.

MGRF Br, sagittal plane moment around the center of mass due to ground reaction forces during the braking phase; MGRF Pr, sagittal plane moment around the center of mass due to ground reaction forces during the propulsive phase; $ROM_T$, trunk range of angular motion; $INC_T$, mean trunk inclination angle; COM, mean center of mass location during stance; DIS, diagonal dissociation; $T_F$, time taken to reach peak vertical GRF in the forelimb; $T_H$, time taken to reach peak vertical GRF in the hindlimb; $P\text{-}R_F$, mean protraction-retraction angle of the forelimb during stance; $P\text{-}R_H$, mean protraction-retraction angle of the hindlimb during stance; COP, mean center of pressure location during stance; $GRFV_F$, peak vertical ground reaction force in the forelimb; $GRFV_H$, peak vertical ground reaction force in the hind limb.

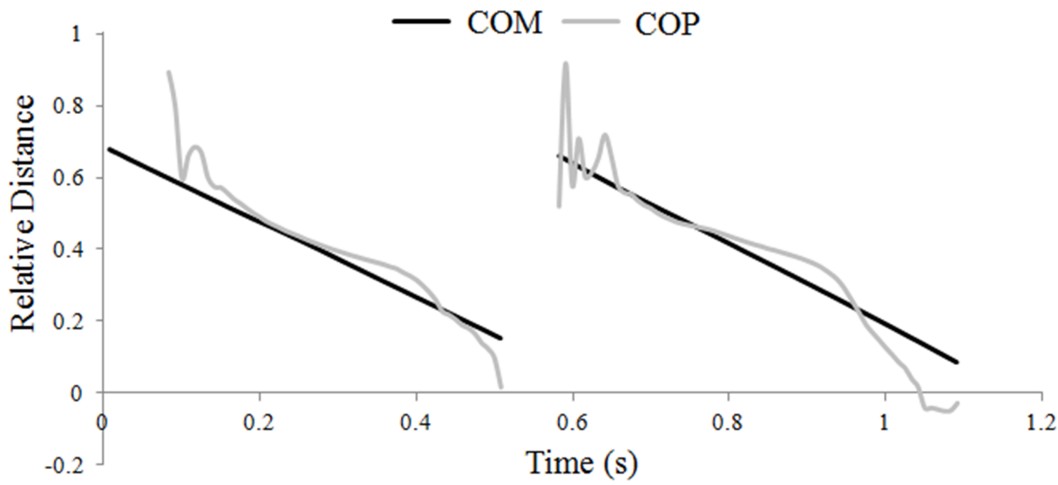

**Figure 1** **Translations of the horse's center of mass (COM) and center of pressure (COP) locations during one stride of passage.** Progression of the COM and COP are shown relative to the grounded fore-limb and expressed as a fraction of the diagonal distance.

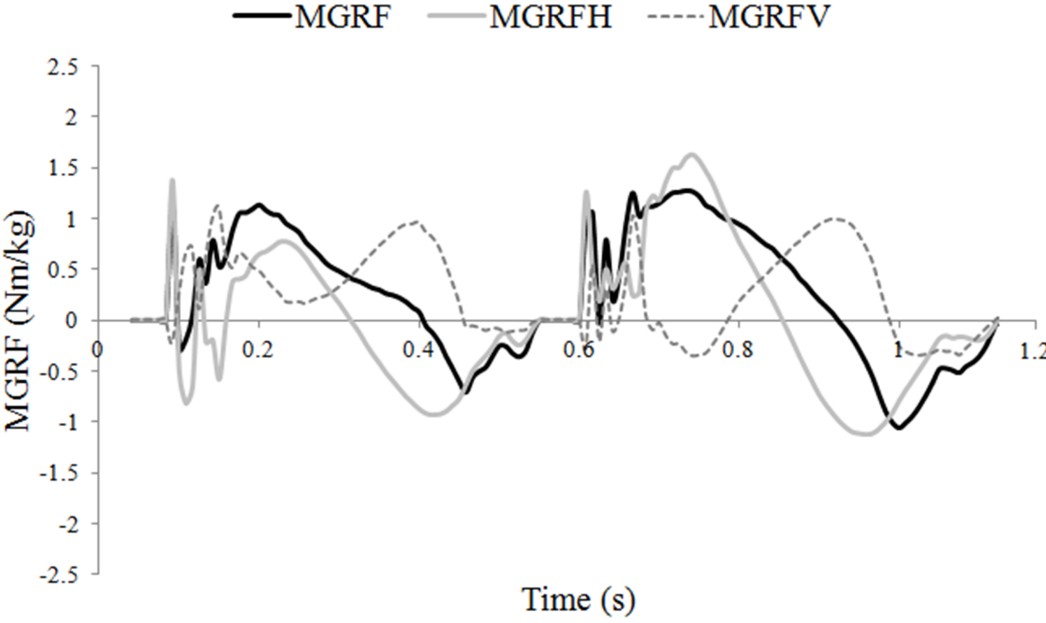

**Figure 2** **Moments around the center of mass (COM) (Nm/kg) during one stride of passage.** MGRF, to-tal moments around the COM due to the effect of the ground reaction forces; $MGRF_V$, moments due to vertical force production; $MGRF_L$, moments due to longitudinal force production.

between limb protraction–retraction angles and peak longitudinal forces to further explore the influence of hoof placement on balance. A strong relationship was found between a more retracted mean angle of the hindlimb and peak hindlimb longitudinal propulsive ($R = 0.821$; $P < .01$) and braking ($R = 0.654$; $P < .01$) forces. There was also a significant relationship between having a less protracted mean forelimb angle and a higher peak forelimb propulsive force ($R = 0.581$; $P < .01$).

## DISCUSSION

Dressage training develops the horse's ability to move with good posture, which involves maintaining the trunk in an uphill (nose-up) orientation, and minimizing the sagittal plane trunk rotations. Under these conditions the horse is described as being in good balance and self-carriage (*Fédération Equestre Internationale, 2014*). The study described here has advanced our understanding of how dressage horses achieve these objectives by identifying kinematic and kinetic variables that are associated with elevation of the forehand and reduction of pitching moments around the COM in passage. The results also indicate areas in which horses that fail to learn the passage might be deficient in strength or coordination.

Dressage doctrine indicates the desirability of the hind hoof stepping further forward relative to the trunk segment in all gaits. In passage less protraction and less retraction are reported in the hind limb compared with trot (*Holmström, Fredricson & Drevemo, 1995*), but the mean hindlimb protraction–retraction angle reported here is similar to that for horses trotting in hand (*Hobbs, Bertram & Clayton, 2016*). This indicates that the hind hoof does not step or remain further forwards in passage compared to trot. Our findings do, however, support the importance of foot contact position, the second motor control strategy, for adjusting moments around the COM in passage by showing that, if hindlimb protraction is limited, for example by conformation, the horse will have difficulty performing passage. Adjustment of limb protraction–retraction has been used to control velocity and maintain fore-aft stability in trotting quadrupedal robots (*Raibert, 1986*; *Raibert, 1990*) and more recently as an additional mechanism to successfully negotiate uneven ground at high speed trot (*Palmer & Orin, 2007*). In robots though, these limb adjustments are made to ensure that synchronous diagonal contacts are achieved between fore and hindlimbs, which controls pitch and roll moments (*Palmer & Orin, 2007*). Quadrupedal mammals are not constrained in this way, as diagonal dissociation, the first motor control strategy may be used to maintain stability during trotting (*Hobbs, Bertram & Clayton, 2016*). In passage, the majority of diagonal contacts were hind-first, but they appeared to play a relatively minor role in managing sagittal plane moments. Subtle differences in contact timing between fore and hindlimbs have been reported previously in dogs (*Lee, Bertram & Todhunter, 1999*), but these were associated with morphological and mass distribution differences between the breeds that were tested (Greyhounds and Labrador Retrievers). In horses, diagonal dissociation has been shown to change the inter-limb timing of force production during stance, which affects the relative GRF contributions of the fore and hind limbs (*Hobbs, Bertram & Clayton, 2016*). In passage, peak forelimb GRF occurred earlier when hind-first dissociation was greater, but this did not influence the relative GRF contributions either. Instead, evidence suggested that hind-first dissociation influenced trunk and limb posture. A greater elevation of the forehand, as indicated by a trunk angle closer to zero, greater hindlimb protraction and an increased trunk ROM were found when a longer time elapsed between hind and forelimb contacts. Greater elevation of the forehand also occurs with greater forelimb protraction. Forelimb posture may dictate trunk pitch on hindlimb landing, after which loading and trunk pitch then influence

hindlimb compression (*Deng et al., 2010*). The overall effect is that an earlier hindlimb landing and greater trunk pitch increase the trunk ROM. This may be a key demand on the hindlimbs when performing passage.

For a horse moving at constant speed, longitudinal GRFs and COM moments over an entire stride should sum to zero. Pitching moments are due to the effects of the GRFs acting at a distance from the COM and to the inertial effects associated with movements of the body segments (*Hobbs, Richards & Clayton, 2014*). With regard to GRF moments, hindlimb vertical GRF and longitudinal braking GRFs create nose-down moments, whereas forelimb vertical GRF and longitudinal propulsive GRFs create nose-up moments. The moment arms of the vertical force components depend on the longitudinal proximity of the hoof to the COM. The moment arms of the longitudinal GRFs are related to limb lengths. Greater hindlimb protraction places the hind hoof closer to the COM, thereby reducing the moment arm of its vertical GRF.

Trotting dogs have been shown to balance the sagittal plane moments by adjusting the fore and hind limb GRFs (*Lee, Bertram & Todhunter, 1999*). In trotting horses, hindlimb vertical GRF is the main contributor to a nose-down moment (*Hobbs, Richards & Clayton, 2014*) but this was not the case in passage in which pitching was controlled by manipulating both the vertical and longitudinal GRF moments (Fig. 2). Hindlimb propulsive force, which contributes to the nose-up moment, is larger in passage than collected trot (*Clayton, Schamhardt & Hobbs, 2017*) and is correlated with increased hindlimb retraction. The ability to use the longitudinal GRF to generate a larger nose-up moment during hindlimb retraction offers a mechanism to combat the increase in nose-down moment associated with the higher hindlimb vertical GRF. Thus, horses may also use the third motor control strategy of adjusting the GRFs to control moments around the COM.

Positioning the hindlimb closer to the COM might be expected to increase its weight-bearing responsibility (*Holmström, Fredricson & Drevemo, 1994*) but, in fact, hindlimb protraction–retraction was not related to hindlimb peak vertical GRF. In contrast, mean forelimb angulation is more protracted in passage than collected trot (*Weishaupt et al., 2009*), which positions the forelimb further away from the COM and forelimb angulation was correlated with a higher peak vertical GRF in the hindlimbs. So the increased weight-bearing responsibility of the hindlimbs is effected by positioning the forelimbs further from the COM throughout the stance phase.

In trotting horses, COP position changes only a little during the middle part of stance with the largest movement occurring in horses that show hind-first diagonal dissociation (*Hobbs, Bertram & Clayton, 2016*). At the same time, the COM is moving forward at almost constant speed and, consequently, the position of the COP relative to the COM changes continuously through stance (*Hobbs & Clayton, 2013*). In passage, the COP tracks the COM much more closely as a consequence of continual adjustments in the vertical GRF ratio between the diagonal limbs such that the relative contribution of the forelimb increases as stance progresses. This causes the COP to move closer to the forelimb at a rate similar to the forward progression of the COM (*Hobbs, Bertram & Clayton, 2016*). When the COP follows the COM more closely, it decreases the moment arm lengths of the vertical GRF, which may be a strategy to reduce moments around the COM. A similar technique is used

by trotting dogs during moderate acceleration and deceleration (*Lee, Bertram & Todhunter, 1999*). Their study found that during acceleration the limbs act in a more retracted position that favours the development of longitudinal propulsive forces, and during deceleration the limbs are held in a more protracted position to facilitate the application of braking forces. The skewed limb positions help to align the resultant GRF vector so it passes close to the COM, thereby reducing the effective moment arm length and, therefore, moments around the COM.

In galloping horses the function of the leading, more protracted forelimb is to provide an impulse that deflects the COM from forwards to upwards (*Bertram & Gutmann, 2009*). Greater mean protraction of the forelimb in passage is likely required to deflect the COM upwards and, together with increased vertical impulses (*Holmström, Fredricson & Drevemo, 1995*; *Clayton, Schamhardt & Hobbs, 2017*), produce a larger vertical COM excursion. Unlike the gallop, this limb posture is not associated with an increase in braking impulse compared to collected trot (*Clayton, Schamhardt & Hobbs, 2017*), but this may be because the forward velocity is slower in passage.

It is difficult to identify owners who are willing to make top quality horses available for research and the opportunity to work with horses of this calibre using these techniques is unusual, though the small number of horses is acknowledged as a limitation to the present study. All horses were of the same breed and ridden by the same rider. Although this may be considered a limitation, a previous study (*Clayton, Schamhardt & Hobbs, 2017*) reported no significant differences in the GRFs generated in passage by Lusitano horses versus Dutch warmblood horses, suggesting that passage is performed similarly across breeds. Finally, the control of balance is not limited to the sagittal plane, but our study did not take into account the influence of altered kinetics and kinematics on medio-lateral balance. Medio-lateral GRF are reported to be highly variable during trotting (*Merkens et al., 1993*), so trunk roll and yaw may be more sensitive to the changes reported here and will be an interesting topic for future study.

## CONCLUSIONS

The mean pitching moments around the horse's COM were managed differently in passage than during trotting, although evidence of all three balancing strategies were found in both gaits. In passage, both the timing (control strategy 1) and position (control strategy 2) of limb contacts play a key role. Adjustments in the fore- and hindlimb vertical GRF (control strategy 3), together with a slower forward COM velocity allowed the COP to track the COM more closely in passage than in trot. Since this reduced the effective moment arm lengths of the vertical GRFs, pitching moments were particularly sensitive to vertical force magnitudes. In trotting, the largest contributor to the nose-down moment is the hindlimb vertical GRF. In passage, the effect of a relative increase in hindlimb vertical GRF in creating a nose-down moment was somewhat countered by either; a more protracted hindlimb position which shortened its moment arm, or by an increase in hindlimb longitudinal propulsive GRF which increased the nose-up moment. Given the complexity of the kinematic and kinetic adjustments required to control pitching moments in passage, it is

not surprising that some horses fail to learn the biomechanical skills necessary to perform this movement well.

### Funding
This work was supported by the McPhail endowment at Michigan State University. The funders had no role in study design, data collection and analysis, decision to publish, or preparation of the manuscript.

### Grant Disclosures
The following grant information was disclosed by the authors:
McPhail endowment at Michigan State University.

### Competing Interests
The authors declare there are no competing interests.

### Author Contributions
- Hilary M. Clayton conceived and designed the experiments, performed the experiments, analyzed the data, contributed reagents/materials/analysis tools, wrote the paper, prepared figures and/or tables, reviewed drafts of the paper.
- Sarah Jane Hobbs conceived and designed the experiments, analyzed the data, contributed reagents/materials/analysis tools, wrote the paper, prepared figures and/or tables, reviewed drafts of the paper, prepared additional data.

### Animal Ethics
The following information was supplied relating to ethical approvals (i.e., approving body and any reference numbers):

The study was performed with approval from the Michigan State University Institutional Animal Care and Use Committee under protocol number 02/08-020-00.

### Data Availability
The raw data has been uploaded as Supplemental Files.

### Supplemental Information
Supplemental information for this article can be found online at http://dx.doi.org/10.7717/peerj.3866#supplemental-information.

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
