# Peer review of "An exploration of strategies used by dressage horses to control moments around the center of mass when performing passage"

_PeerJ, doi:10.7717/peerj.3866_

## Round 0.1 · original submission · Minor Revisions

Congratulations-- two reviewers, both kindly non-anonymous, have supplied constructive reviews for this MS, and agree that it is publishable with only minor to moderate revisions. I do not anticipate that further review will be required, but please ensure that the Response with the revised MS addresses all points raised by the reviewers so that these can be checked and a final decision made.

·

Basic reporting

No comment in general, please add abbreviations for table in legend

Experimental design

No comment

Validity of the findings

No comment

Additional comments

This paper studies the biomechanical variables which influence pitching moments around the COM in passage in elite dressage horses. It uses comprehensive recording and analysis to understand the background for the high biomechanical skills of three elite dressage horses.
There are two points regarding the study object and presentation of the results which should be considered:
1. “balance control”: In the abstract, introduction and discussion the term balance or balance control, balance management is used. You also defined this term in the abstract as the ability to control nose up/nose down pitching moments. This might be a quite limited part of the general locomotor balance of the horse. To my knowledge the FEI does not exactly define this term “balance”. While this study focusses on the sagittal-plane motion, the transversal plane movements might be of even more importance especially during slow gaits. As balance might mean generally “a state of equilibrium“ and the capacity to maintain postural stability, it might be nice to discuss these aspects in your discussion in more detail.
2. Results: Table 1: Please provide the explanations for the abbreviations in the table legend.

·

Basic reporting

Please see the annotated PDF for minor comments on phrasing.

I find there are limited references within the article, particularly in the introduction. It is not mentioned that this topic is understudied, so I am surprised that there are not more citations. I don’t believe in including irrelevant citations for the sake of it, but some mention of limited references, or inclusion of more support for discussions of trot and pitching moments in other movements may be beneficial. This would also help to demonstrate how this research fits with the broader field.

It would be more logical to define pitching moments in the introduction, as it is an important aspect of the study, as opposed to the discussion (Lines 219-221).

The figures are clear and appropriate to the article; however, I’m not sure why they are subplots of the same figure. It would be clearer to have two distinct figures, and these should also be referred to specifically in the text. Figure 1 is referred to, but not 1A or 1B.

Experimental design

There are minor phrasing comments within the annotated PDF to clarify the Materials and Methods section, but overall it contains sufficient detail.

The research fits within the Aims and Scope of the journal as stated on the website, and the research question is fairly well defined. The statement “some dressage horses fail to reach the highest levels…” (Lines 65-67) is interesting and a nice rationale for the study, but is buried within the introduction.

The authors use the term “elite” as well as “highly-trained” dressage horses. I think there needs to be a clear definition of these terms. The use of only three horses seems very small for “highly-trained” dressage horses, but it more acceptable if “elite” is referring only to Olympic competitors. There should also be acknowledgement of the small sample size before the end of the discussion, which could be accomplished by defining the criteria for inclusion.

Validity of the findings

I believe the methods and analysis are sound, and the results presented are reliable. I particularly like the supplementary video.

The discussion and conclusions are appropriate to the results of this study, but there are aspects of the discussion that I think could be improved.

The first sentence of the discussion states that dressage “…involves maintaining the trunk in an uphill (nose-up) orientation” (Line 201). This is a key factor of the pitching moments discussed in the paper, but is not mentioned before the discussion. This should be brought to the attention of the reader in the introduction, and would also assist with identifying the purpose of the study.

The manuscript overall is focused on healthy horses performing a difficult movement; the mention of disease (Line 211) seems off topic and unnecessary. I think it should be removed.

I don’t think it’s necessary to mention that there are no significant differences between breeds when all the individuals were the same breed (lines 262-265).

The final sentence of the conclusions refers back to lines 65-67, which I have already stated is where I believe the key rationale for this study to be. But, as it’s not mentioned within the discussion, this last sentence seems oddly placed. I recommend highlighting this more within the introduction and discussion.

The discussion overall could be brought together more effectively. It reads a little disjointed, each result is discussed separately, rather than passage overall. There could be greater comparison between passage and trot, as this is briefly mentioned but not clearly compared. For example, lines 227-229, the comparison here is not clear (I’m not sure what “controlled by manipulating both […] GRF moments” means in relation to the trot).

There is also little mention of the three balance strategies that are highlighted earlier in the manuscript. They are included in Table 1, but then not discussed. This would be a good way to bring together all the information from the discussion to an overall mechanism for balance in the passage.

Additional comments

Overall, I think this is a well written, PeerJ appropriate manuscript. The methods are ethical and appropriate, the data is sound, and raw data is provided as supplementary information. There are areas that could be improved to demonstrate the research question that this manuscript is aiming to address.

---

## Round 0.2 · Minor Revisions

I have checked the revision and response and am satisfied with the changes except on one point: as per the reviewer's comment on pitching moments around the COM, the MS focuses on horses and mainly the authors' own work. Indeed with just 10 references most are by the authors. The Intro (and ideally Discussion) needs a broader discussion of how this work fits into broader understanding of pitching moments in other animals. A literature search on this topic would surely uncover work on humans, birds, horses, dogs, and other animals (and robots)- e.g. https://goo.gl/4BYNBy, and the value of this paper would be considerably expanded if it was thought of in that broader context instead of the narrower focus of dressage horses. Please make according amendments and resubmit - thank you for your patience.

---

## Round 0.3 · accepted · Accept

I have checked the Tracked Changes and am convinced that reasonable diligence has been done following my recommendations from the prior draft; with the paper's value to a broader readership (and connection to broader science, too) being increased. Thank you for making those changes and congratulations on having the paper accepted!